# A Novel Marine Ranching Cages Positioning system on Unmanned Surface Vehicles Using LiDAR and Monocular Camera Fusion

Jiewen Li

Naval Architecture and Shipping College
Guangdong Ocean University
Zhanjiang, China
lijiewen_gd@163.com

Qiao Liu

Naval Architecture and Shipping College
Guangdong Ocean University
Zhanjiang, China
qiaoliu@gdou.edu.cn

Ronghui Li

Naval Architecture and Shipping College
Guangdong Ocean University
Zhanjiang, China
Lironghui@163.com

Jiayi Lai

Naval Architecture and Shipping College
Guangdong Ocean University
Zhanjiang, China
ljyoranges@163.com

*Abstract*—This paper introduces a novel positioning system designed to enhance the positioning accuracy of unmanned surface vehicles (USVs) during feeding operations near marine ranching cages. The system integrates monocular camera and LiDAR data through a tolerance-based matching algorithm to achieve precise positioning. Initially, real-time environmental images are captured by the camera, and object detection is performed on these images using the YOLOv8 algorithm, which facilitates the extraction of bounding boxes and coordinates for preliminary positioning. Simultaneously, the LiDAR point cloud data are preprocessed and then clustered with the DBSCAN algorithm to derive accurate distance and angle measurements. Subsequently, a tolerance-based matching algorithm is employed to fuse the LiDAR and camera data, leveraging precise distance and angle thresholds to optimize data alignment. Additionally, Real-time visualization of the fused data is achieved with the ROS rviz tool, providing a comprehensive view of target positions and enabling detailed monitoring and analysis. Experiments conducted on land using simulated marine ranching cages validate the feasibility and effectiveness of the system, demonstrating its robustness and reliability through repeated testing.

*Index Terms*—Surface Unmanned Vessel(USV), Environment Sensing, Data Fusion, Target Positioning

## I. INTRODUCTION

Marine ranching has emerged as a pivotal strategy for the sustainable management of marine resources, significantly enhancing the efficiency and productivity of aquatic farming. This method involves the deployment of cages in natural marine environments, optimizing spatial utilization while minimizing ecological impacts, as illustrated in figure 1, which depicts the Zhanjiang marine ranching site. Despite its benefits, marine ranching requires regular and

Project funding and support were provided in part by the National Natural Science Foundation of China (grant number:52171346) the special projects of key fields of Universities in Guangdong Province (grant number:2023ZDZX3003)

precise feeding to ensure optimal fish growth and health. Traditional manual feeding methods involve significant labor and high operational costs. In contrast, unmanned surface vessels (USVs) offer substantial advantages for autonomous feeding, including reduced fuel consumption, lighter hulls, and lower manpower requirements, thereby improving cost-effectiveness and efficiency.

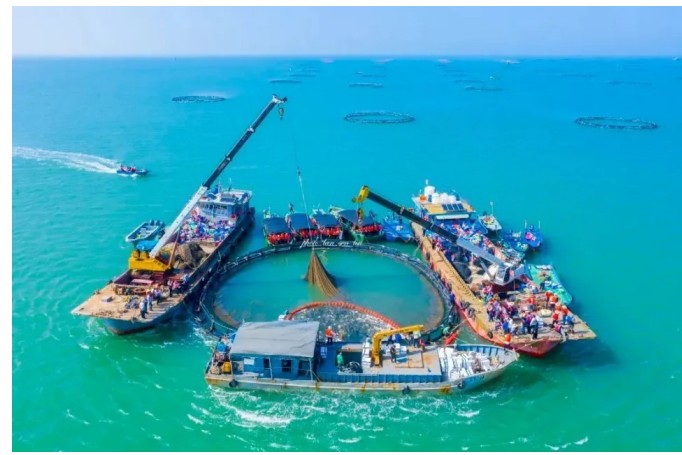

Fig. 1. zhanjiang's marine ranching

The effective deployment of autonomous feeding by USVs depends critically on the real-time positioning of cages. Accurate positioning is required to ensure that feed is delivered precisely to the intended areas, thereby optimizing feeding efficiency and promoting optimal fish health. Existing positioning methods, such as GPS-based systems and basic visual tracking, face limitations in complex marine environments. GPS signals can be compromised by interference or occlusion, and visual tracking

systems are affected by lighting variations and restricted field of view. These limitations become particularly pronounced when USVs operate at close proximity to cages, where inaccurate positioning may lead to collisions with the cages, potentially causing damage to the infrastructure and endangering the health of the fish. Thus, there is an urgent need for a precise positioning system specifically designed for near-field cage feeding operations to ensure safe and efficient autonomous feeding.

Most USVs are equipped with both cameras and LiDAR for environmental perception. Cameras provide detailed RGB information for target recognition but are constrained by their two-dimensional nature and sensitivity to lighting conditions, affecting position accuracy. Conversely, LiDAR offers precise three-dimensional point cloud data, which is less impacted by occlusion and lighting conditions but lacks color information and produces sparse data. Combining the three-dimensional spatial data from LiDAR with the color and detail information from cameras enables the system to achieve higher accuracy in environmental perception. The integration of LiDAR and camera data addresses the shortcomings of each individual sensor.

Given the limitations of traditional positioning methods and the advantages offered by sensor fusion, this study proposes a novel system for cage positioning using the combined data from LiDAR and monocular cameras. The objective is to leverage multi-sensor fusion to achieve precise and reliable cage positioning in challenging near-field conditions, enhancing autonomous feeding accuracy and mitigating collision risks to safeguard infrastructure and fish within the marine ranching site.

## II. RELATED WORK

### A. Camera-Based Target Positioning

Camera-based target positioning has been extensively studied due to its importance in applications such as autonomous driving, surveillance, and robotics. These methods generally involve estimating distances through depth cues and camera parameters, with recent advancements being driven by the application of deep learning techniques for object detection.

The R-CNN series [1]–[3], including Fast R-CNN and Faster R-CNN, have significantly improved detection accuracy by integrating region proposals with convolutional networks. For instance, Hu et al. [4] employed Faster R-CNN for distance and angle estimation, demonstrating its effectiveness across various environments. However, these approaches require substantial computational resources, limiting their real-time applicability, particularly in high frame rate scenarios.

The YOLO (You Only Look Once) series, especially from YOLOv3 to YOLOv8 [5]–[9], are favored for their real-time processing capabilities. Studies by Natanael et al. [10] and Haseeb et al. [11] showed how YOLO could be used for distance estimation and long-range obstacle detection with monocular cameras. Additionally, enhancements like those by Mauri et al. [7] and Dist-YOLO [8] improve distance estimation but increase model complexity, posing challenges in resource-constrained deployments.

Despite these advancements, camera-based methods alone are limited by the inherent constraints of 2D data, particularly the lack of depth information critical for accurate positioning in dynamic, three-dimensional environments. These limitations highlight the need for multi-sensor fusion approaches that combine camera data with depth information from sensors like LiDAR.

### B. LiDAR-Camera Fusion Methods

The integration of LiDAR and camera data has gained traction due to the complementary strengths of these sensors. LiDAR offers precise 3D spatial data, which is less impacted by lighting and occlusions, while cameras provide detailed color and texture information. Fusion techniques are broadly categorized into result-level, proposal-level, and point-level methods.

Result-Level Fusion generates 3D proposals based on 2D camera detections, as seen in FPointNet [12] and RoarNet [13]. However, these methods often struggle with sensor misalignment and do not fully leverage the contextual information from both sensors.

Proposal-Level Fusion, used in methods like MV3D [14] and AVOD [15], combines data at the region proposal stage. However, they can be adversely affected by significant background noise in rectangular regions of interest (RoIs), leading to suboptimal performance in cluttered environments.

Point-Level Fusion, demonstrated by approaches like Point-Painting [16], shows promise by directly associating point cloud data with image features. However, these methods are still susceptible to sensor misalignment and may not fully capture the contextual relationships between data points, particularly in dynamic or complex environments.

While these fusion methods offer improvements over single-sensor approaches, they still face challenges in terms of robustness and accuracy in real-time applications, especially when dealing with the dynamic and often unpredictable conditions found in marine environments. Sensor misalignment, the complexity of data integration, and the need for real-time processing remain significant obstacles.

## III. SYSTEM DESIGN

This study proposes a robust multi-sensor fusion system that improves real-time positioning and operational efficiency in complex environments by integrating LiDAR and monocular camera data with a tolerance-based matching algorithm. The system's architecture consists of several key components, each meticulously developed to ensure efficient sensor data fusion and reliable environmental

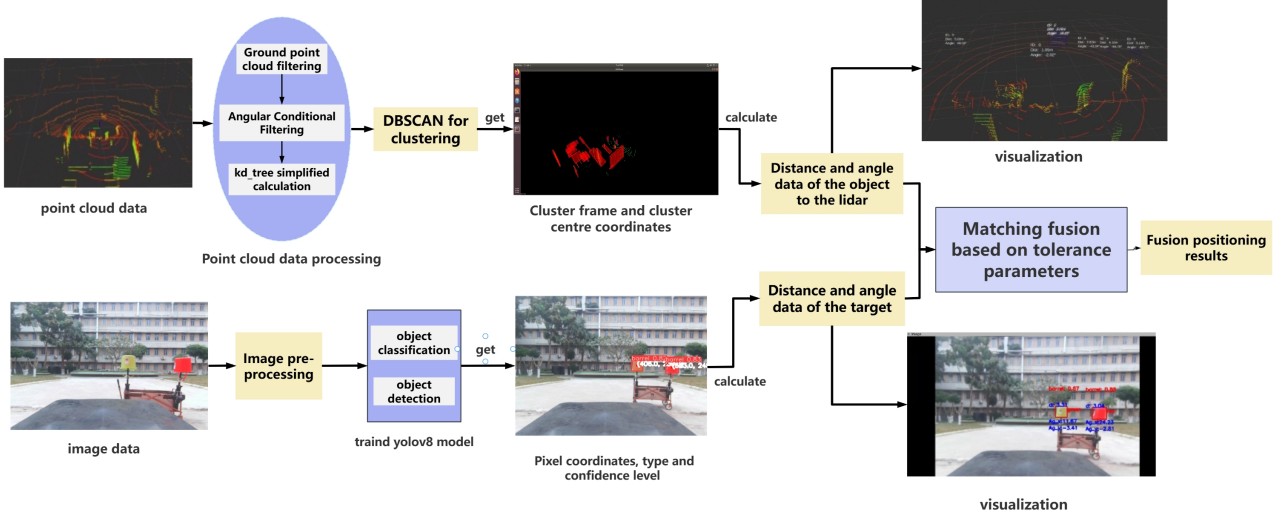

Fig. 2. Fusion System Flowchart

perception. Inputs are processed from two primary sensors: the camera and LiDAR. Visual data is captured by the camera, and object detection is performed using the YOLOv8 algorithm, which extracts precise bounding boxes and coordinates. The depth information is collected by LiDAR through point clouds, and object positions are determined after filtering out ground points and applying the DBSCAN algorithm to cluster the point cloud. These data are then fused in the data fusion module to generate the target's position and orientation, effectively compensating for the limitations inherent in each sensor. The workflow of the system is outlined as follows:

Step 1: System Initialization – ROS nodes are configured for the camera and LiDAR sensors.

Step 2: Camera Data Processing – Environmental images are captured by the camera and processed using the YOLOv8 model to detect and localize objects.

Step 3: LiDAR Data Processing – Point cloud data are collected, filtered, and clustered to obtain positioning information for multiple objects.

Step 4: Data Fusion – The data from the camera and LiDAR are combined to achieve comprehensive positioning information.

Step 5: Output and Visualization – The fused data are visualized in the ROS environment for real-time analysis and decision-making.

The system flow chart is provided in figure 2. This structured and modular approach ensures that high-precision and reliable perception data are delivered, even in complex and dynamic environments. In summary, the integration of camera and LiDAR data has been effectively designed to enhance the accuracy of target detection and positioning. The following sections proceed with a detailed discussion of the subsystem methodology.

## IV. CAMERA-BASED SYSTEM

### A. Monocular camera calibration

To ensure accurate correspondence between spatial and image points, a geometric model is established to project 3D objects onto the 2D image plane. Due to lens distortions and external factors, calibration is required to correct these distortions. This involves determining the Intrinsic Parameters, Extrinsic Parameters, and Distortion Coefficients of the camera. Tsai's algorithm [17] and Zhang Zhengyou's calibration algorithm [18] are commonly used for this purpose. In this study, the camera_calibration tool within the ROS framework, using a checkerboard pattern was employed for monocular camera calibration.

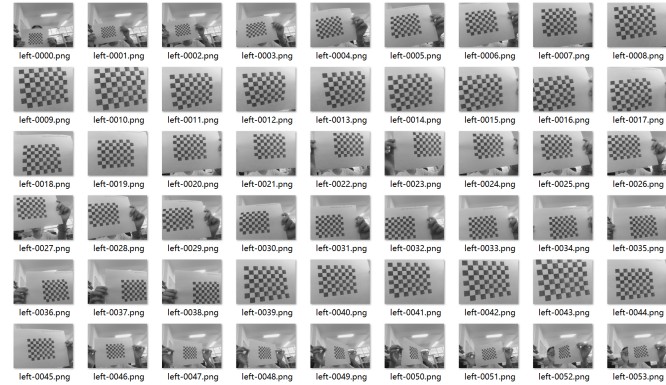

Fig. 3. Camera intrinsics calibration process.

Figure 3 illustrates the camera intrinsics calibration process. The transformation from world coordinates to image coordinates begins with converting the world coordinate system to the camera coordinate system. Here, $C$ denotes the camera origin, and $O$ the world origin. This transformation involves a rotation matrix $R$ and a

translation vector $t$. The transformation is expressed as: $\tilde{X}cam = R(\tilde{X} - \tilde{C})$, which in matrix form is:

$$tX_{cam} = \begin{bmatrix} \tilde{X}_{cam} \\ 1 \end{bmatrix} = \begin{bmatrix} R & -R\tilde{C} \\ 0 & 1 \end{bmatrix} \begin{bmatrix} \tilde{X} \\ 1 \end{bmatrix} = \begin{bmatrix} R & -R\tilde{C} \\ 0 & 1 \end{bmatrix} X \tag{1}$$

The camera coordinate system is defined as $(X_c, Y_c, Z_c)$, where $Z_c$ represents the optical axis. The image plane is positioned at a distance $f$ from the camera along the optical axis direction, where $f$ denotes the focal length. The camera intrinsics describe the transformation from pixel coordinates to image coordinates, including the focal length, optical center, and skew factor. The camera intrinsics matrix $C_i$ is represented as:

$$C_i = \begin{bmatrix} f_x & s & c_x \\ 0 & f_y & c_y \\ 0 & 0 & 1 \end{bmatrix} \tag{2}$$

This calibration process is crucial for ensuring accurate mapping between the 3D world and the 2D image plane, serving as the foundation for subsequent tasks such as precise object detection, localization, and multi-sensor data fusion that will be conducted in this study.

### B. Object Detection Using YOLOv8

Cameras play a pivotal role in object detection and recognition by capturing detailed visual information. This study employs the YOLOv8 model [9], an advanced iteration in the YOLO series, known for its fast and accurate real-time detection capabilities. YOLOv8 enhances traditional methods by partitioning the image into grids and predicting both the class and bounding box for each object, thereby enabling simultaneous detection of multiple objects and improving system performance.

In this study, the dataset was utilized to train the YOLOv8 neural network, with the training conducted on a high-performance GPU to optimize the model for real-time detection tasks. The training process was carried out iteratively, using a large batch of labeled images, until satisfactory accuracy was achieved on the validation set. After 150 epochs of training, a significant improvement in target detection accuracy was observed. Subsequently, the trained neural network weights were integrated into the constructed model [19], enabling real-time detection and recognition. Input images were processed by the YOLOv8 network, resulting in visual outputs that included target bounding boxes and category probabilities.

As illustrated in figure 4, a target is detected at a distance of 3 meters from the camera, positioned at a 20° angle to the right. Each detected target is labeled with its corresponding confidence level, demonstrating that the YOLOv8 model achieves accurate and reliable target detection.

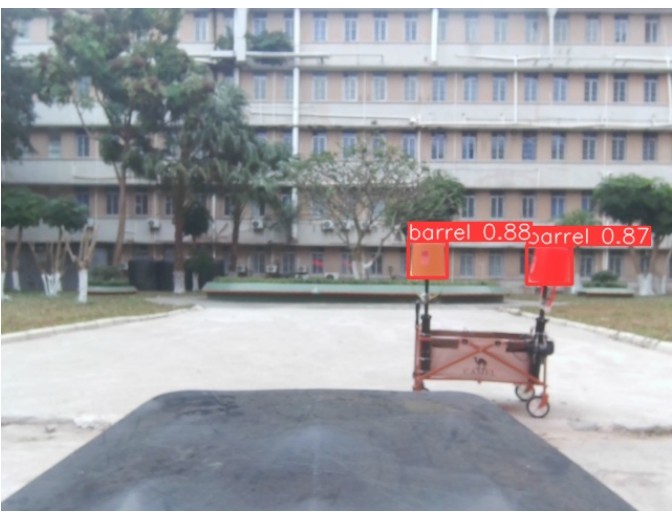

Fig. 4. Target detection results.

### C. Camera-Based Distance and Angle Estimation

After detecting objects and obtaining their coordinates, the camera estimates the distance and angle to each object. This process involves calculating the spatial relationship based on bounding boxes and pixel coordinates, using triangulation techniques for precise measurements.

1) Target Ranging: A monocular vision-based method estimates the distance from the camera to the target by measuring the target's bounding box and using known camera parameters, such as focal length and optical center, obtained through calibration. The distance is calculated using:

$$\text{Distance} = \frac{F \cdot H \cdot h_i}{h_b \cdot H_s} \tag{3}$$

where $h_i$ is the total pixel height of the image.

2) Target Azimuth: The target's angle relative to the camera center is calculated by converting image coordinates to camera coordinates. The pixel deviation from the image center is determined:

$$\delta x = x - c_x, \quad \delta y = y - c_y \tag{4}$$

This deviation is then used to compute the angle:

$$\theta_x = \arctan\left(\frac{\delta x}{f_x}\right), \quad \theta_y = \arctan\left(\frac{\delta y}{f_y}\right) \tag{5}$$

Finally, the angles are converted to degrees:

$$\theta_x^{\deg} = \theta_x \times \left(\frac{180}{\pi}\right), \quad \theta_y^{\deg} = \theta_y \times \left(\frac{180}{\pi}\right) \tag{6}$$

### V. LiDAR-BASED SYSTEM

Simultaneously, the LiDAR sensor performs environmental scans and collects point cloud data. Before clustering, the point cloud data undergoes preprocessing to filter out noise and irrelevant points. Clustering algorithms

are then applied to the refined data to identify distinct objects and determine their cluster centers. Each cluster center is interpreted as the position of an object within the horizontal plane, thereby providing multiple sets of distance and angle measurements for all detected objects.

### A. Point cloud data preprocessing

The large number of point clouds in a frame of 3D LiDAR data results in a more discrete distribution and complex spatial structure. To enhance data processing speed, it is crucial to reduce the point cloud volume by eliminating unnecessary data and retaining only the relevant target points.

*1) Ground Point Cloud Filtering:* Ground point cloud data, considered as noise for target positioning, is filtered out. For an unmanned vessel operating on a flat surface, ground data are fitted using the Randomized Analysis and Consistency of Surface Sampling (RANSAC) algorithm [20]. RANSAC is a simple, iterative method that accurately fits mathematical model parameters from a dataset containing outliers [21].

*2) Angular Conditional Filtering:* Conditional filtering is applied to select or exclude points based on geometric or attribute conditions. In this study, only points in the front-facing region of the LiDAR are retained. The LiDAR records the position and angle of each point, which are used to determine its position relative to the front of the device. Points are traversed and checked against the defined frontal angle range. Points meeting the criteria are retained for fusion with camera target data [22].

### B. Point cloud DBSCAN clustering

The DBSCAN algorithm is highly effective for clustering point cloud data, particularly in cases of uniform density and small inter-point distances. It excels over methods like K-means by not requiring a predefined number of clusters and handling arbitrary-shaped data. Despite its higher computational complexity, DBSCAN's ability to identify clusters without prior cluster count makes it ideal for LiDAR point cloud applications [23].

As shown in figure 5, DBSCAN groups point clouds into clusters or identifies noise points. The angle $\beta$ between points A and B, relative to the laser radar origin O, is calculated using:

$$\beta = \arctan\left(\frac{|BC|}{|AC|}\right) \quad = \arctan\left(\frac{|OB|\sin\alpha}{|OA| - |OB|\cos\alpha}\right) \tag{7}$$

If $\beta > \theta$, points A and B are considered to be at the same depth; if $\beta < \theta$, they are deemed to be in different clusters.

### C. LiDAR-based positioning

*1) Cluster Bounding Box Calculation:* After clustering the point cloud data, Axis-Aligned Bounding Boxes (AABBs) are generated for each cluster. The AABB is

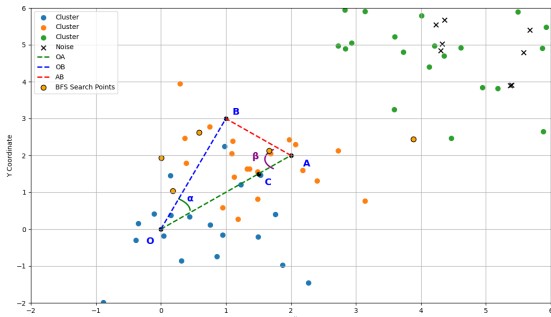

Fig. 5. DBSCAN clustering applied to point cloud data.

determined by calculating the minimum and maximum coordinates of all points within the cluster, using the formula:

$$AABB = [min(x), max(x); min(y), max(y); min(z), max(z)] \tag{8}$$

This approach provides a straightforward way to estimate the object's position and size, which can then be visualized using 3D tools.

*2) Bounding Box Center Calculation:* The geometric center of the bounding box is calculated to determine the centroid of the cluster, using:

$$\text{Center} = \left(\frac{x_{\min} + x_{\max}}{2}, \frac{y_{\min} + y_{\max}}{2}, \frac{z_{\min} + z_{\max}}{2}\right) \tag{9}$$

This centroid represents the average spatial position of the cluster and is useful for visualizing the cluster's shape and orientation.

*3) Distance Calculation from Cluster Center to LIDAR:* The distance from the cluster center to the LIDAR sensor is computed using the Euclidean distance formula:

$$\text{Distance} = \sqrt{(x - x_0)^2 + (y - y_0)^2 + (z - z_0)^2} \tag{10}$$

Here, $(x_0, y_0, z_0)$ represent the LIDAR sensor's position, typically at the origin $(0, 0, 0)$. This calculation provides the straight-line distance from the cluster's center to the sensor.

*4) Cluster Center Azimuth Calculation:* The azimuth angle, crucial for target localization, is calculated using the atan2 function:

$$\theta = \arctan 2(y, x) \tag{11}$$

This function accurately determines the angle by considering the signs of $x$ and $y$. The angle is then converted from radians to degrees, ensuring precise measurement.

## VI. DATA FUSIOM SYSTEM

### A. Camera and Lidar Time Synchronization

In multi-sensor systems integrating cameras and LiDAR, addressing frequency discrepancies is crucial for accurate data fusion. The camera operates at 30 Hz, while the LiDAR at 10 Hz, leading to potential misalignment without proper synchronization. To address this, a hard trigger method is employed, where the LiDAR triggers the camera to ensure temporal alignment.

This study utilizes the Robot Operating System (ROS) for precise time synchronization using message filters. These filters cache and synchronize incoming sensor data based on specific conditions, ensuring alignment without altering the data.

Figure 6 illustrates the impact of synchronization on camera and LiDAR data integration. Panel (a) shows unsynchronized timelines with misaligned data, while Panel (b) displays synchronized timelines with accurate data correlation, emphasizing the importance of synchronization for robust and accurate multi-sensor data fusion.

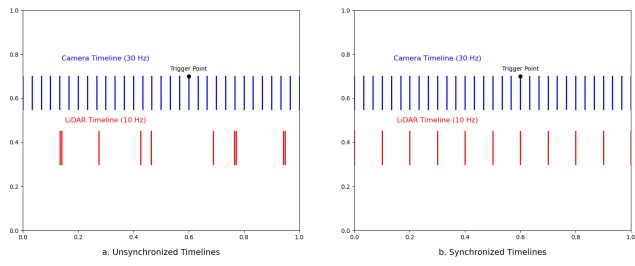

Fig. 6. Synchronization of Camera and LiDAR Data

### B. Tolerance-based Matching Algorithm

Efficient and accurate data matching is crucial for enhancing multi-sensor fusion systems. This paper introduces a tolerance-based matching algorithm for aligning LiDAR and camera data using specific distance and angle tolerances. The algorithm reduces sensor noise and improves detection reliability by comparing the estimated distances and angles of objects from both sensors.

1) Tolerance parameter: These parameters define the maximum allowable differences during matching:

(1) Distance tolerance (*dist_tolerance*): Specifies the maximum acceptable difference in distance between data points.

(2) Angle Tolerance (*angle_tolerance*): Specifies the maximum acceptable difference in angle between data points.

2) Matching Logic: The matching criteria are defined as follows:

$$\text{distance diff} = |d_{lidar} - d_{camera}|$$
$$\text{angle diff} = |\theta_{lidar} - \theta_{camera}| \tag{12}$$

Where d denotes the distance and $\theta$ denotes the angle, two data points are considered matched if the following conditions are met:

$$\text{distance\_diff} \leq \text{dist\_tolerance}$$
$$\text{angle\_diff} \leq \text{angle\_tolerance} \tag{13}$$

Figure 7 illustrates the matching process using tolerance-based criteria, where red points represent LiDAR data and blue points represent camera data. The dashed circles indicate the allowable distance tolerance, while the dashed lines connect pairs of data points that satisfy both distance and angle criteria. This figure

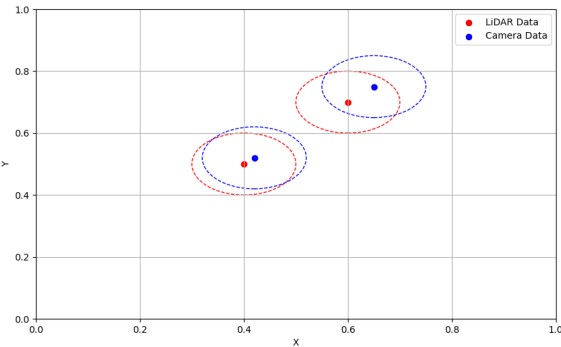

Fig. 7. Tolerance-Based Data Matching

effectively demonstrates the successful alignment of multi-sensor data, highlighting the robustness of the proposed method in practical applications. Such alignment is crucial for ensuring accurate and reliable data fusion, particularly in complex operational environments.

## VII. EXPERIMENTAL RESULTS

This experiment matches and fuses the data after processing the LiDAR point cloud and camera image separately to achieve more accurate target positioning, enabling the USV to better identify specific targets.

### A. Experimental platforms

The experiments were conducted on an Ubuntu 20.04 operating system with ROS Noetic. The visualization of real-time point cloud and image positioning information was achieved using RViz. The flexibility and modularity of the ROS platform allowed multiple nodes to run and be debugged simultaneously, effectively meeting the complex needs of practical application scenarios. The sensor parameters are as follows:

(1) Monocular Camera: This experiment uses the camera model XFK 4K415. The maximum resolution of the camera is 3840 * 2160, the frame rate is 30fps, through the USB2.0 interface for data transmission and power supply, reliable data, while the camera supports MJPG, YUYV image output format. The lens chosen is 3.2mm distortion-free and the lens shooting angle is 105°.

(2) LIDAR: This experiment uses robosense LIDAR Helios16, the number of lines is 16 lines, the ranging capability is 150m, the horizontal angle of view is 360°, the vertical field of view is 30° (-15° to 15°), and the content of the UDP packet is the three-dimensional spatial coordinates, the reflective intensity, and the timestamp.

## B. Data Preparation and Processing

1) Camera Data Preparation: The camera data used in this experiment were collected on land using a ROS system, which recorded rosbags from USV-mounted cameras as the targets were positioned at different angles and distances. The data acquisition environment is shown in figure 8. JPG images were extracted from the collected rosbag and utilized as the dataset for this experiment. A total of 2,540 JPG images were selected for this dataset, of which 2,286 were used for training and 254 for testing. The images were annotated separately using the LabelMe tool and the annotations were converted into a dataset format compatible with YOLOv8.

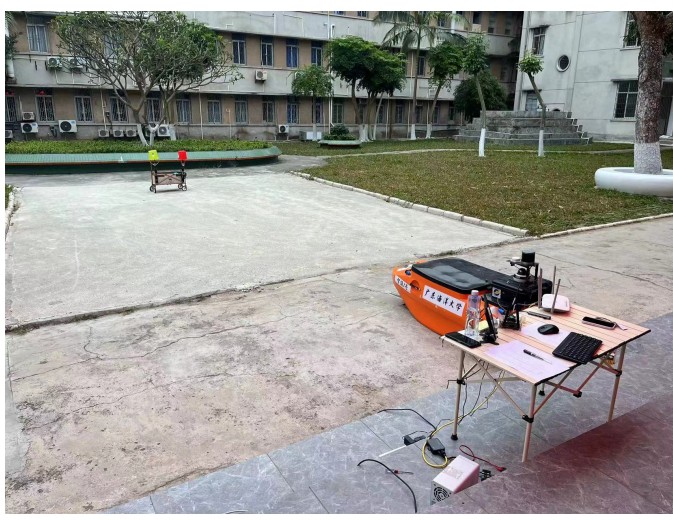

Fig. 8.   Dataset shooting environment.

2) LiDAR Data Collection and Visualization: The experimental point cloud data used in this experiment is homemade with the ROS system recorded on the land unmanned ship mounted LiDAR with the target of a number of groups of different angles and distances rosbag, the shooting environment as shown in figure 8, the rosbag package can be achieved in the visualization of the real-time observation of the point cloud data in the visualization of the rviz, the visualization of the figure 9.

## C. Visualization of Experimental Results

The processed LiDAR point cloud and camera images, each enriched with precise target information, were visualized in ROS's rviz. These enhanced visualizations provide detailed views of target positions, which are essential for optimizing the USV's autonomous navigation and feeding operations.

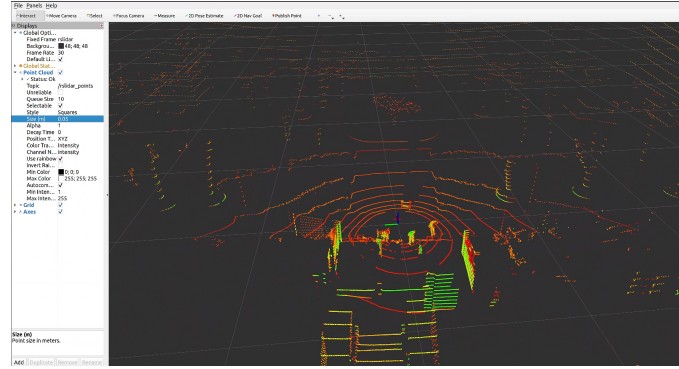

Fig. 9.   Visualizing point clouds.

1) Camera positioning Information: Image processing nodes were executed to record the specific distance and angle of the target using the monocular camera. The localization information was obtained through target detection and positioning algorithms, with visualization performed using RViz. The results are presented in figure 10.

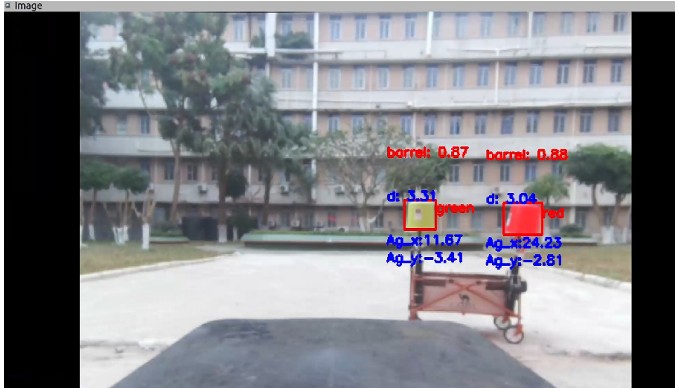

Fig. 10.   Camera positioning visualization.

The figure illustrates the target's category, confidence level, color, distance from the camera, and horizontal and vertical angles relative to the camera.

2) LiDAR Positioning Information: The LiDAR processing ROS node was executed, utilizing the recorded rosbag containing data at specific distances and angles to generate real-time position information through LiDAR clustering. This environmental position data was then visualized using RViz, as illustrated in figure 11.

The figure demonstrates the distance and angle between multiple clustering results and the sensors, enabling accurate LiDAR positioning.

3) Fusion of LiDAR and Camera Data: Multiple processing nodes were executed simultaneously, outputting their respective target positioning information. The positioning data from both sensors were visualized in RViz, as depicted in figure 12.

4) Sensor Data Matching: The tolerance-based data matching algorithm was implemented on the Robot Op-

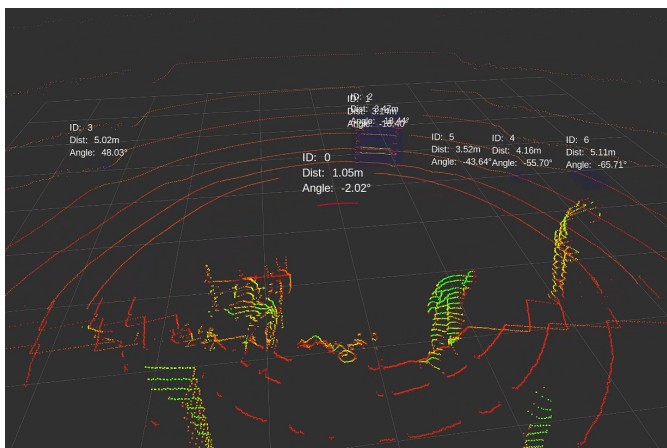

Fig. 11. Lidar positioning visualization.

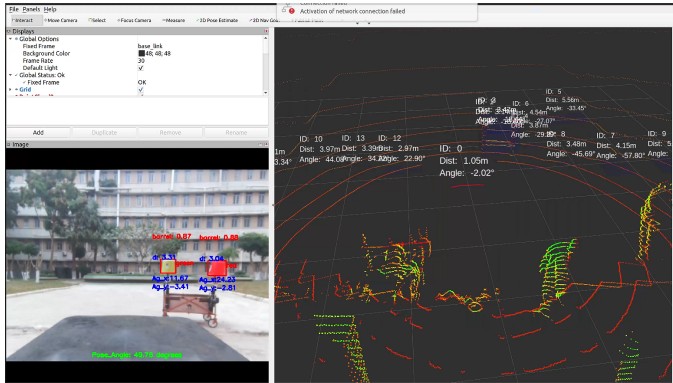

Fig. 12. Dual Sensor Positioning Visualization.

erating System (ROS) platform. The experiment concurrently received the processed position information from both the camera and LiDAR, matching the data using the tolerance-based algorithm and promptly outputting the results in real time. When the target was positioned directly 2 meters in front of the unmanned vessel, the output was recorded as depicted in figure 13.

```
[INFO] [1716796735.492193]: Fused Data - Distance: 2.22, Angle: 0.23
target:[(2.06, -9.53), (2.19, 8.79)]
[INFO] [1716796736.491345]: Fused Data - Distance: 2.22, Angle: 0.23
```

Fig. 13. Data matching results.

### D. Analysis of Results

To verify the effectiveness of the proposed algorithm, this experiment utilized 10 sets of rosbag data collected simultaneously by LIDAR and the camera at various angles and distances. The target's distance and azimuth were calculated through data processing and fusion matching, with results compared to the actual position, as detailed in Table I.

As indicated in Table I, a discernible trend is observed where the error in distance estimation increases with

TABLE I
Fusion Results for Multiple Data Sets

| Physical Location | Distance (m) | Angle (deg) | Distance Error (m) | Angle Error (deg) |
|---|---|---|---|---|
| 0200 | 2.22 | 0.23 | 0.22 | 0.23 |
| 0230 | 2.23 | 29.73 | 0.23 | 0.27 |
| 0300 | 3.15 | 0.49 | 0.15 | 0.49 |
| 0320 | 3.28 | 19.60 | 0.28 | 0.40 |
| 0410 | 4.31 | 9.36 | 0.31 | 0.64 |
| 0420 | 4.24 | 19.31 | 0.24 | 0.69 |
| 0510 | 5.35 | 10.73 | 0.35 | 0.73 |
| a0210 | 2.22 | -10.08 | 0.22 | 0.08 |
| a0310 | 3.25 | -10.75 | 0.25 | 0.75 |
| a0420 | 4.26 | -20.97 | 0.26 | 0.97 |

the target's distance from the sensor. For example, at approximately 2 meters, the error is relatively small, ranging from 0.22 to 0.23 meters. In contrast, at a distance of 5.35 meters, the error rises to 0.35 meters. This suggests that as the distance increases, the accuracy of the positioning system decreases.

Several factors contribute to this trend. The sensor's inherent resolution limitation, which worsens with distance, is a primary factor. For LIDAR and camera systems, angular resolution becomes less effective at greater distances, compounding measurement errors. Furthermore, discrepancies may arise from aligning data from sensors with differing resolutions and perspectives, particularly at the sensors' range limits.

Despite these challenges, the multi-sensor fusion system demonstrates strong performance in precise target localization within close-range environments, crucial for near-field autonomous operations like marine ranching feeding. The small error margins at shorter distances highlight the system's capability for accurate positioning in such scenarios.

In conclusion, while the system's error increases with distance, its high precision in close-range applications underscores its suitability for environments requiring accurate positioning. The implementation of a tolerance-based matching strategy effectively integrates multi-source sensor data, ensuring reliable performance even under complex conditions.

## VIII. CONCLUSIONS

In this paper, a novel multi-sensor fusion system has been proposed to enhance the precision of cage positioning for unmanned surface vehicles (USVs) in marine ranching environments. Utilizing a tolerance-based matching algorithm to integrate LiDAR and camera data, the system effectively overcomes the limitations of traditional GPS and visual tracking methods, achieving high precision in target positioning.

Experimental validations have demonstrated that positioning accuracy diminishes with increasing distance due to inherent constraints in sensor resolution and angular effectiveness. Nonetheless, the system has proven to be highly effective in close-range scenarios, with small error margins at shorter distances underscoring its capability to provide accurate positioning in confined environments. This precision is particularly beneficial for applications

requiring high accuracy in near-field settings, such as automated feeding in marine ranching cages.

This research not only showcases the potential of multi-sensor fusion in improving target positioning accuracy in close-range environments but also lays the groundwork for future studies. Future research should focus on expanding the system's capabilities by integrating additional sensors and refining algorithms to further reduce error margins. Additionally, it is essential to validate the system's effectiveness and scalability in real-world marine conditions, with particular emphasis on its performance across varying distances and environmental factors.

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
