# OpenReview forum: "A Novel Marine Ranching Cages Positioning system on Unmanned Surface Vehicles Using LiDAR and Monocular Camera Fusion"
_IEEE.org/ICIST/2024/Conference — IEEE ICIST 2024 Conference Submission_

### Official Review · Reviewer_PtSy · 2024-08-21
**Good paper**

**Rating:** 7
**Confidence:** 3

**Review:**

This paper introduces a novel positioning system designed to enhance the positioning accuracy of unmanned
surface vehicles (USVs) during feeding operations near marine ranching cages.The content of this article is abundant and innovative, and it is recommended for acceptance.  I would provide the following three constructive suggestions for further improvement:
1. While the experiments conducted on land using simulated marine ranching ages have demonstrated the feasibility of the system, it would be highly beneficial to conduct comprehensive field tests in real-world marine environments.  Real-world testing can reveal challenges unique to the marine setting, such as wave action, changes in lighting conditions, and underwater obstructions.  This will enable the researchers to further refine and optimize the system for practical applications.

2. To increase the system's robustness and reliability, consider integrating additional sensors such as inertial measurement units (IMUs) or depth sensors.  These additional sensors can provide supplementary data that can be fused with the LiDAR and camera data, reducing reliance on any single sensor and enhancing the overall system's fault tolerance.

3. The article mentions that positioning accuracy diminishes with increasing distance.  To expand the system's operational range and maintain high accuracy over longer distances, it would be valuable to investigate and implement advanced algorithms that can compensate for the inherent limitations of LiDAR and camera sensors at greater distances.  This could involve developing machine learning models specifically trained for long-range distance and angle estimation or optimizing existing clustering and matching algorithms for better performance at further distances.

---

### Official Review · Reviewer_Gg5J · 2024-08-21
**This paper introduces a novel positioning system designed to enhance the positioning accuracy of unmanned surface vehicles (USVs) during feeding operations near marine ranching cages. The feasibility of the designed control approach is proven via the simulation example. However, the following suggestions need to be considered in the revised manuscript to further improve the quality of this paper.**

**Rating:** 7
**Confidence:** 3

**Review:**

1. How does the proposed sensor fusion system improve positioning accuracy compared to traditional methods?
2. Can you provide more details on the tolerance-based matching algorithm used to fuse the LiDAR and camera data?
3. The format of the reference is problematic. For instance, it is suggested to change the “a” in reference [20] to “An”. There are many similar errors in this paper. Please standardize all incorrectly formatted references to the correct format.

---

### Official Review · Reviewer_ccAR · 2024-08-21
**This paper is well-organized, has convincing simulation results, and is recommended for publication.**

**Rating:** 7
**Confidence:** 3

**Review:**

This paper presents a certain progress in the field of unmanned surface vehicle (USV) positioning, particularly during feeding operations near marine ranching cages. The innovative system integrates data from a monocular camera and LiDAR through a tolerance-based matching algorithm, achieving a notable improvement in positioning accuracy. The use of YOLOv8 for object detection and DBSCAN for LiDAR data clustering ensures that both visual and spatial data are processed effectively to provide accurate preliminary positioning and precise distance and angle measurements. The reviewer has the following questions to discuss with the authors:

1. How does the tolerance-based matching algorithm specifically optimize the alignment of LiDAR and camera data? What challenges were encountered in fusing these two data sources, and how were they addressed?

2. Why was the YOLOv8 algorithm chosen for object detection in this system? How does its performance compare to other object detection algorithms in the context of marine environments?

3. What specific metrics or criteria were used to assess the robustness and reliability of the positioning system during the experiments? Were there any scenarios where the system struggled or needed further optimization?

---

### Decision · Program_Chairs · 2024-09-08

Accept (Oral)